# Response to FEC Chemotherapy and Oncolytic HSV-1 Is Associated with Macrophage Polarization and Increased Expression of S100A8/A9 in Triple Negative Breast Cancer

**DOI:** 10.3390/cancers13215590

**Published:** 2021-11-08

**Authors:** Alyssa Vito, Nader El-Sayes, Omar Salem, Yonghong Wan, Karen L. Mossman

**Affiliations:** McMaster Immunology Research Centre, Department of Medicine, McMaster University, Hamilton, ON L8S 4K1, Canada; vitoar@mcmaster.ca (A.V.); elsayesn@mcmaster.ca (N.E.-S.); salemo@mcmaster.ca (O.S.); wanyong@mcmaster.ca (Y.W.)

**Keywords:** triple-negative breast cancer, breast cancer, immunotherapy, B cells, myeloid cells, tumor microenvironment, oncolytic virotherapy, chemotherapy

## Abstract

**Simple Summary:**

We have previously reported that a combination of clinical chemotherapies and oncolytic HSV-1 works to sensitize tumors to respond to immune checkpoint blockade. We further showed that this therapeutic platform worked via the upregulation of B cells and the concomitant control of immunosuppressive myeloid cells. In this manuscript, we sought to further dissect the mechanism of myeloid cell regulation and differentiation and to identify a therapeutically driven gene signature that is associated with the switch in the myeloid phenotype. This work not only impacts triple-negative breast cancer but all solid tumor phenotypes as we aim to better understand the underlying immunology associated with responses to immune checkpoint therapies in these typically refractory disease types.

**Abstract:**

The era of immunotherapy has seen an insurgence of novel therapies driving oncologic research and the clinical management of the disease. We have previously reported that a combination of chemotherapy (FEC) and oncolytic virotherapy (oHSV-1) can be used to sensitize otherwise non-responsive tumors to immune checkpoint blockade and that tumor-infiltrating B cells are required for the efficacy of our therapeutic regimen in a murine model of triple-negative breast cancer. In the studies herein, we have performed gene expression profiling using microarray analyses and have investigated the differential gene expression between tumors treated with FEC + oHSV-1 versus untreated tumors. In this work, we uncovered a therapeutically driven switch of the myeloid phenotype and a gene signature driving increased tumor cell killing.

## 1. Introduction

In recent years, we have seen an insurgence of novel immunotherapies in both preclinical and clinical development, changing the face of cancer therapy and the clinical management of the disease. However, the percentage of patients that benefit from these highly efficacious therapies is low, and recent focus has shifted towards the need for a more comprehensive understanding of the immune system and immune interactions within the ever-changing tumor microenvironment (TME) [1]. With renewed attention to fundamental tumor immunology, myeloid cells have been presented as a vital and fluid population of cells that drive immune response, promote the activation and expansion of effector T cells, and also simultaneously play a role in the maintenance of tissue homeostasis and in promoting immune tolerance [2]. Tumor-infiltrating myeloid cells are found abundantly in the tumor stroma, and their levels strongly correlate to patient outcomes in many forms of cancer [3]. Additionally, tumors can co-opt myeloid cells to promote cancer growth and increase metastatic potential [3]. Recent years have uncovered a crucial role of myeloid cell-driven immune escape in the TME, and it has become widely recognized that myeloid cells play a central but not fully understood role in the response to many types of therapy [4].

In cancer, a decrease in peripheral myeloid cells drives myelopoiesis, increasing the migration of these cells before they have completely differentiated [5]. This results in an influx of myeloid cells with strong immunosuppressive patterns and abnormal functions [6,7]. Due to their myeloid origin, this heterogenous population of cells has been termed myeloid-derived suppressor cells (MDSCs), representing a distinct population of immature myeloid cells that are activated under sustained inflammation. In breast cancer, MDSCs have been demonstrated as a major driver of immune escape and the main reason for therapeutic resistance and cancer relapse, particularly in the case of immunotherapy [4,8]. Circulating MDSCs in the peripheral blood of breast cancer patients are elevated in all stages of the disease and are directly correlated with clinical cancer stage and metastatic burden [9,10]. While MDSCs are a major source of immunosuppression within the TME, tumor-associated macrophages (TAMs) also represent a distinct suppressive population of cells that is known to drive tumorigenesis and metastasis [11].

We have previously reported an immunotherapy platform targeting immune-bare triple-negative breast cancer (TNBC) tumors using a clinical chemotherapy cocktail (FEC; 5-fluorouracil, epirubicin, cyclophosphamide) in combination with an oncolytic Herpes Simplex virus type 1 (oHSV-1) [12]. In this work, we showed that FEC + oHSV-1 therapy sensitized tumors to immune checkpoint blockade (ICB) and the requirements of tumor-infiltrating B cells to combat MDSC-driven immunosuppression in the TME. Herein, we have further investigated these effects and have focused on the transcriptomic changes of FEC + oHSV-1 therapy to switch the genomic landscape of a murine TNBC tumor from that of immunosuppressive myeloid phenotypes to those with antitumorigenic functions. In particular, we looked at macrophage frequencies, as previous data support a role for B cells in mediating macrophage polarization [13,14,15].

Preliminary data for this work were conducted in subcutaneous tumors, which fail to accurately recapitulate de novo tumor formation and lack other cell types that may be found in naturally occurring TMEs. To assess the locational differences between subcutaneous tumors and those in the more appropriate location of the mammary fat pad, genome-wide transcriptome analysis was performed in both subcutaneous and orthotopically implanted tumors. While these studies are specific to breast cancer, the underlying immunological functions can carry through to other solid tumor phenotypes.

## 2. Materials and Methods

### 2.1. Cell Lines

Human osteosarcoma cells (U2OS; ATCC, Manassas, VA, USA) were used for virus preparation. U2OS cells were maintained in Dulbecco’s modified Eagle’s media (DMEM) and were supplemented with 10% fetal bovine serum (FBS; ATCC 30-2020), 2 mmol/L L-glutamine, 100 U/mL penicillin, and 100 μg/mL streptomycin (Gibco, Grand Island, NY, USA). Murine breast adenocarcinoma cells were isolated from a spontaneous tumor in a C57/Bl6 mouse (E0771; CH3 Biosystems, Amherst, NY, USA) and were used for in vivo experiments. E0771 cells were maintained in Roswell Park Memorial Institute (RPMI) medium supplemented with 10% FBS and 2 mmol/L l-glutamine. All of the cell lines used in our experiments were grown in an incubator that was maintained at 37 °C with 5% CO_2_.

### 2.2. Mouse Experiments

All of the mouse experiments were completed in the McMaster University Central Animal Facility. All procedures using mice were conducted in compliance with the Canadian Council on Animal Care and under the approval of the Animal Research Ethics Board of McMaster University (Animal Utilization Protocol 17-05-22). Six- to eight-week-old female C57/Bl6 mice (Charles River Laboratories, Wilmington, MA) were used in all of the mouse experiments. For subcutaneous the E0771 tumors, 5 × 10^6^ cells in 200 µL PBS were implanted on the left flank. For orthotopic tumors, 2 × 10^5^ cells in 50 µL PBS were implanted on the lower-left mammary fat pad. For all of the experiments, the mice were housed 5/cage, fed a normal diet, and kept at room temperature. After tumor implantation, it took approximately twelve days for palpable tumors to arise (50–100 mm^3^). All of the mice were randomized before treatment commenced. FEC was given on day 1 (20 mg/kg 5-fluorouracil in 200 μL saline, followed by 3 mg/kg epirubicin in 200 μL saline, followed by 20 mg/kg cyclophosphamide in 200 μL saline 1 h later; intraperitoneally). oHSV-1 was given on days 2, 3, and 4 (2 × 10^7^ pfu oHSV-1 dICP0 in 50 μL PBS, intratumorally). Tumors were measured every 2 days with calipers, and tumor volumes were calculated as volume = length × width2/2. For the survival studies, the mice were classified as reaching endpoint when their tumor volume reached 1000 mm^3^, when they lost 20% of their original body weight, or when they had a body score of 1.

### 2.3. Chemotherapy Treatments

All chemotherapies were stored as per the manufacturer’s recommendation and were resuspended to the desired concentration in sterile saline on the same day that they were to be used [5-fluorouracil, Sigma Aldrich (St. Louis, MO, USA), F6627, 2 mg/mL; Epirubicin, Cayman Chemicals, 12,091, 0.3 mg/mL; cyclophosphamide, Sigma Aldrich, C0768, 2 mg/mL].

### 2.4. Oncolytic Virus Treatments

Oncolytic HSV-1 was produced by a homologous recombination using infectious DNA of luciferase-expression wild-type HSV-1 KOS/Dluc/oriL [16]. HSV-1 dICP0 contains a deletion of the entire *ICP0* coding region. oHSV-1 was propagated and titered on U2OS cells with 3 mmol/L hexamethylene bisacetamide (Sigma) and were purified via sucrose cushion ultracentrifugation. The final purified virus was resuspended in PBS to the desired concentration and were stored in aliquots at −80 °C for experimental use.

### 2.5. Survival Analysis: Kaplan–Meier and ROC Plots

The publicly available online software KMplot.com was used to generate the Kaplan–Meier survival plots in Figure 3 [17]. This database contains breast cancer microarray datasets derived from multiple publicly available datasets available on the NCBI Gene Expression Omnibus (GEO). The use of this tool is described in detail by the developers in Győrffy et al. 2010 [17]. Probes were selected using JetSet optimization, and the average expression of the macrophage signature (Table 1) was analyzed using mRNA expression logarithmic values above (high) or below (low) either the median or upper tertile for relapse-free survival. ROCplot.org was used to assess the prognostic significance of S100A8 and S100A9 (Figure 7) [18]. Patient microarrays with treatment and response annotation were accrued by the developers using the NCBI GEO [18]. They defined a pathologically complete response as being a pathologically complete response versus residual disease after completing therapy. S100A8 and S100A9 mRNA expression were independently assessed as predictors of a pathologically complete response in any patients treated with combined 5-fluorouracil, epirubicin, and cyclophosphamide (FEC) therapy, and the mRNA expression of S100A8 and S100A9 was plotted for responders versus non-responders using Graphpad Prism.

### 2.6. Clariom S Assay

RNA was extracted from whole tumor digests using the Qiagen RNA extraction kit (Cat #74004). Extracted RNA was diluted to a concentration of 100 ng/50 µL and underwent reverse transcription. sscDNA was purified using magnetic beads and fragmented using UDG. The fragmented sample was hybridized to the Affymetrix Clariom S mouse arrays, and the stained arrays were scanned to generate intensity data. All of the reagents for this assay were developed by and purchased from Thermo Fisher Scientific (Loughborough, UK). Raw data was analyzed using the Thermo Fisher Transcriptome Analysis Console software. The complete dataset can be found in the GEO database, GSE183864.

### 2.7. Flow Cytometry

Cells were extracted from spleens by pressing them between two glass slides, and blood was collected from the periorbital sinus. Red blood cells from all of the samples were lysed using ACK buffer. The PBMCs were treated with Fc block (anti-CD16/CD32; BD Biosciences (Mississauga, Ont., Canada), #553141) and surface stained antibodies for FVS (BD Biosciences, #564406), CD19 (Fisher Scientific, #14-019-482), B220 (BD Biosciences, #563894), CD4 (BD Biosciences, #561830), CD8 (BD Biosciences, #563046), CD11b (BD Biosciences, #553311), Ly6C (BD Biosciences, #553104), Ly6G (BD Biosciences, #560602), F4/80 (BD Biosciences, #743282), and S100A8/A9 (Novus Biologicals, #NBP2-47667AF700). The LSRFortessa flow cytometer was used to run samples, and FACSDiva software (BD Biosciences) was used for data acquisition. FlowJo software was used to analyze the flow cytometry data.

### 2.8. Statistical Analysis

One-way analysis and *t* test were used to determine the statistical significance of the differences in the means between groups. The log-rank (Mantel–Cox) test was used to analyze the statistical significance between the groups in the Kaplan–Meier survival graphs. All statistics were conducted using GraphPad Prism (La Jolla, CA, USA).

## 3. Results

### 3.1. FEC + oHSV-1 Improves Survival Outcomes

C57/Bl6 mice were implanted with E0771 tumors subcutaneously on the left flank and were treated with either PBS, a combination of clinical chemotherapies (FEC; 5-fluorouracil, epirubicin and cyclophosphamide), oncolytic HSV-1 (oHSV-1), or FEC + oHSV-1. While neither therapy showed as much of an effect on tumor growth as the monotherapy approach did, the mice treated with the dual combination of FEC + oHSV-1 had delayed tumor progression and 10–20% durable responses to treatment (Figure 1). These data are consistent with our previously published findings using this therapeutic combination [12].

### 3.2. FEC + oHSV-1 Upregulates Inflammatory Myeloid Cell Pathways

To validate previously published findings from RNA sequencing data, we performed gene expression profiling using the microarray technique. C57/Bl6 mice were implanted with E0771 cells subcutaneously on the left flank and were treated with either PBS (*n* = 5) or FEC + oHSV-1 (*n* = 10). Tumors were harvested on day 5, and RNA was extracted from the whole tumor digests. The principal component analysis of the data shows that mice treated with FEC + oHSV-1 cluster distinctly from those treated with PBS (Figure 2A). The pathway enrichment analysis shows that FEC + oHSV-1 therapy switches the myeloid phenotype in the tumor from that of an immunosuppressive nature to that of an inflammatory one. This switch is characterized by STAT5 and STAT3 signaling with a strong inflammatory response and the upregulation of genes associated with apoptosis. Gene-set enrichment analysis also revealed a strong downregulation of MTORC1 signaling, a known driver of myeloid cell differentiation to the immunosuppressive MDSC phenotype (Figure 2C) [27]. Asper our previous publications, microarray analysis showed that FEC + oHSV-1 therapy increased the genes associated with the B cell receptor signaling pathways. Of the top 100 genes from the B cell signature gene list that we previously published [12], 64 overlapped with the current dataset (Appendix A).

Consistent with FEC + oHSV-1 therapy achieving durable responses in 10–20% of treated mice (Figure 1C and [11]), the transcriptomic profiling of the FEC + oHSV-1-treated mice shows that 80% of the mice clustered more similarly to the PBS-treated mice, with 20% of the mice having a distinct expression profile (Figure 2D). We speculate that the mice with a distinct expression profile correspond to responders to therapy, though the causal relationship cannot be shown, as different cohorts of mice were used for each experiment. Further, to determine whether these results were biased by the subcutaneous nature of the tumors, orthotopic implantation was used to more closely mimic the true microenvironment of breast tumors (Appendix A). Data are consistent between both implantation models, but gene expression differences were more pronounced in the tumors derived from the mammary fat pad.

Of the top upregulated genes (Figure 2B), many are associated with macrophages (*LCN2*, *CXCL2*, *SAA3*, *CXCL3, CCL3*, *IL1A*, *IL1B, CLEC4E*), which are key myeloid-derived cells that are known to play a primary role in epigenetic reprogramming [28,29,30]. To assess the clinical impact of these findings, we assessed the prognostic significance of this macrophage gene signature that was upregulated by FEC + oHSV-1 therapy (Table 1) in a combined cohort of publicly available clinical microarray data [17]. The mean expression of this signature above median was associated with better relapse-free survival (RFS) in TNBC-restricted patient data (HR = 0.49 (0.3–0.82); logrank *p* value = 0.0051) (Figure 3A) and across all breast cancer patients combined (HR = 1.73 (1.32–2.27); logrank *p* value < 0.01) (Figure 3B). These findings suggest that the myeloid cell signature generated by FEC + oHSV-1 therapy could have positive prognostic implications.

### 3.3. FEC + oHSV-1 Increases Inflammatory Myeloid Cells in the Peripheral Blood and Spleen

To assess the changes occurring in the immune cell populations as a result of FEC + oHSV-1 therapy, we performed flow cytometry analysis (Figure 4). The mice were treated with PBS, FEC, oHSV-1, or FEC + oHSV-1, and blood was collected 9 and 13 days after the start of treatment. While there were no significant changes in T cells, B cells, or MDSCs, FEC + oHSV-1 therapy significantly increased the amount of circulating inflammatory monocytes (CD11b^+^Ly6C^hi^Ly6G^-^ cells) and macrophages (F4/80^+^ cells) on both days 9 and 13 (Figure 4).

While classical myelopoiesis occurs in the bone marrow, recent data suggest that the spleen is a prominent site of extramedullary hematopoiesis (EMH) in cancer [31]. For this reason, we decided to look into the spleen of tumor-bearing mice treated with PBS, FEC, oHSV-1, or FEC + oHSV-1. Interestingly, while we found that the level of M1-like macrophages did not change in the peripheral blood (Figure 5A) and that the overall frequency of macrophages was not significantly changed across treatment groups in the spleen, the macrophages isolated from the spleens of the mice treated with FEC + oHSV-1 expressed iNOS, suggesting an M1-like phenotype (Figure 5B,C). This further supports the role of FEC + oHSV-1 therapy in polarizing macrophages to an inflammatory phenotype.

### 3.4. FEC + oHSV-1 Therapeutic Efficacy Further Potentiated by Upregulation of S100A8/A9

We closely investigated the top upregulated genes in the transcriptomic profile of the tumors from mice the treated with FEC + oHSV-1 therapy. Not only was the switch in the myeloid phenotype driven by the genes associated with macrophages, but additionally, two of the top upregulated genes were *S100A8* and *S100A9,* which key players in myeloid cell differentiation and modulation. While the exact mechanism of pro- and antitumorigenic functions of S100A8/A9 remain elusive to date, studies have shown that this heterodimer drives apoptotic pathways when found at high concentrations [32,33,34,35].

Mice bearing E0771 subcutaneous tumors were treated with PBS, FEC, oHSV-1, or FEC + oHSV-1, and flow cytometry analysis was performed on the peripheral blood 6, 9, 10, 13, and 15 days after the start of treatment. FEC + oHSV-1 therapy increased levels of S100A8/A9 in the peripheral blood (Figure 6A). Furthermore, the expression was predominantly localized to the F4/80^+^ macrophages, peaking on day 10 Figure 6B,C and Appendix A). Interestingly, when we looked at the expression of S100A8/A9 on the M1-like macrophages (CD11b + F4/80 + iNOS + S100A8/A9 + cells), we saw an increase in this population with FEC + oHSV-1 therapy (Figure 6D). These findings are in line with the RNA transcriptome profile, further indicating the importance of macrophages in shifting myeloid cell differentiation.

### 3.5. S100A8 and S100A9 Are Predictive of Response to FEC Therapy in Breast Cancer

To further assess the prognostic relevance of elevated levels of S100A8 and S100A9 in breast cancer, we utilized a publicly available cohort of combined clinical microarray data containing response rates to various therapies [18]. The evaluation of S100A8 and S100A9 mRNA expression (independently) in the patients treated with FEC therapy indicated that both S100A8 and S100A9 were predictors of complete pathological response (S100A8: AUC = 0.71, *p* < 0.0001; S100A9: AUC = 0.626, *p* = 0.0014) (Figure 7).

## 4. Discussion

Immunotherapy has continued to cement itself as a pillar of cancer care, with widespread clinical success in a variety of cancer types. However, responses to immunotherapy treatments, and in particular ICB, vary greatly between patients, and the drivers of therapeutic response have remained largely elusive to date. While myeloid cells have arisen as potent regulators of tumor evolution, their plasticity and metabolic heterogeneity means that they are heavily influenced by the TME and cellular populations that they come in contact with [36]. The accumulation of MDSCs in breast tumors is largely driven by cytokine and chemokine production and can influence the differentiation of TAMs to more suppressive phenotypes [37]. Additionally, B cells are known to be drivers of myeloid cell differentiation and key influencers of the cytokine and chemokine secretome in the TME [12,14,15]. While we often talk about cells of the myeloid lineage as being static cell types, it is important to remember that they are dynamic, plastic, and consistently undergoing differentiation. One way in which we can characterize their functional state is by the proteins that they express/secrete at a given time.

S100A8 and S100A9 are calcium-binding proteins that belong to the S100 family. They often exist as a heterodimer and have minimal function in the homodimer state due to instability [32]. This heterodimer is constitutively expressed by myeloid cells, which can function as a calcium sensor, with roles in cytoskeletal rearrangement and metabolic pathways [32,38]. In response to inflammation and cellular stress, S100A8/A9 is released from the cytoplasm and actively participates in the modulation of immune homeostasis by stimulating leukocyte recruitment and by inducing cytokine secretion [32]. While these proteins have been extensively studied across various disease types, their exact role in inflammatory and malignant conditions continues to be controversial in the literature. In particular, S100A8 and S100A9 have been described as having both pro- and antitumorigenic functions [39,40].

We have previously reported a combination of FEC + oHSV-1 therapy as being capable of sensitizing tumors to ICB. To further investigate this phenomenon, herein, we have described the RNA profile of tumors treated with either FEC + oHSV-1 therapy or PBS and the myeloid gene signature associated with our therapeutic platform. Notable from our findings, FEC + oHSV-1 significantly downregulated MTORC1 signaling. Indeed, S100A9 has been shown to control the MTORC1 modulation of MDSCs [41,42,43]. Additionally, recently published studies have also dictated that therapies that can successfully sensitize breast tumors to ICB do so through the epigenetic reprogramming of myeloid cells in the TME, shifting TAMs to an antitumor M1 phenotype and promoting the STAT3-mediated suppression of myeloid cells [44]. These findings are in line with our analysis of cellular populations in PBMCs.

While clinical studies have demonstrated the prognostic relevance of TILs in many subtypes of breast cancer [45,46], TNBC patients are found to have more tumors with intermediate or high levels of TILs than other non-TNBC subtypes of the disease [47]. Increased TILs are associated with a better response to therapy and improved overall survival [46]. The correlative effects of TILs are limited not only to their density in the TME but more notably on the phenotypic state of the infiltrates. In our model, we showed that the level of macrophages and other myeloid-lineage cells may be consistent in terms of the overall frequency, but their phenotype largely determines their functionality and pro- or antitumorigenic functions. Specifically, the polarization of macrophages to an M1 phenotype drives inflammatory myeloid cells and antitumor immunity. Our data suggest that this polarization occurs in the spleen, as the frequency of macrophages exhibiting M1-like characteristics are uniform across all treatment groups when assessed in the blood but rapidly undergo differentiation in the spleen.

These findings are in line with other studies assessing the regulation and importance of myeloid cells in the TME. Takabe and colleagues have shown that TAMs play a crucial role in breast cancer biology and that tumors bearing higher levels of M1 macrophages had a more favorable tumor immune landscape [48]. Interestingly, Gabrilovich et al. have shown that the presence of S100A9-positive macrophages was predictive of a poor clinical outcome in patients with head and neck cancer and of a poor response to ICB in patients with metastatic melanoma [49]. These findings are contradictory to our own studies and suggest that further analysis is required to determine if S100A8/A9 expression as a predictor of therapeutic outcome is cancer-type dependent. Indeed, it may be plausible that the expression levels of S100A8/A9 in our model were elevated as a response to extended cellular damage, stress, and oncolysis in the TME from chemotherapy and oncolytic virotherapy.

It is important to note a limitation of our RNA microarray analysis, as we used whole tumor digests for RNA extraction. In the future, it will be interesting and important to isolate myeloid cells from tumors and to perform RNA analysis on specific cell types (or alternatively use single cell sequencing methods) to identify strong mechanistic links and ensure that the gene signatures that we have extracted from our data can be directly attributed to MDSCs and/or macrophages. Together, our findings combined with others in the literature further highlight the complexity of the immunosuppressive mechanisms and plasticity of myeloid cells within the TME, a phenomenon that warrants further analysis as we aim to improve responses to immunotherapy treatments and to better understand their mechanistic functions.

## Figures and Tables

**Figure 1 cancers-13-05590-f001:**
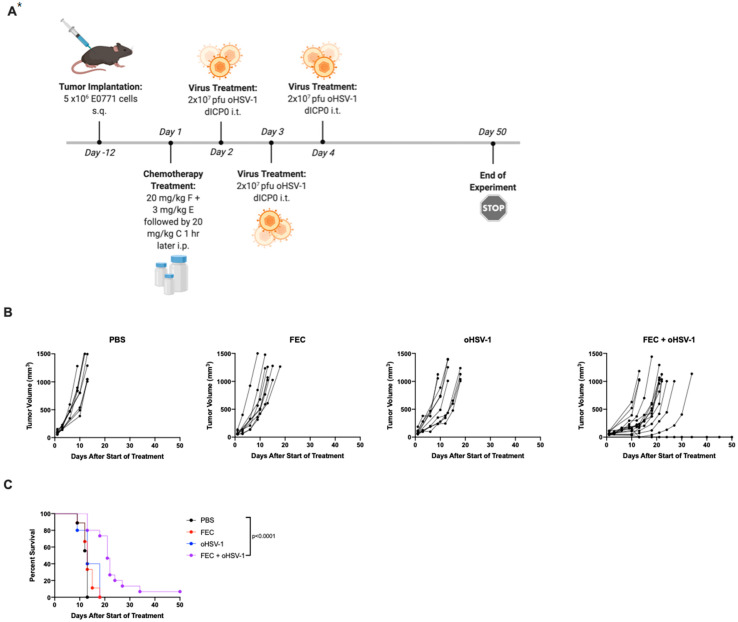
A combination of clinical chemotherapy and oncolytic HSV-1 synergizes for improved disease control in E0771 tumors; (**A**) C57/Bl6 mice bearing E0771 tumors on the left flank were treated with PBS (*n* = 9), chemotherapy (FEC, *n* = 9), oncolytic virus (oHSV-1, *n* = 9), or chemotherapy + oncolytic virus (FEC + oHSV-1, *n* = 14). * Created using BioRender.com. (**B**) Tumor volumes were measured every 2–3 days from the start of treatment until mice reached tumor endpoint (volume = 1000 mm^3^). Each line represents an individual mouse within the group. (**C**) Kaplan–Meier survival curves of each group. * Mantel–Cox test was used for statistical analyses.

**Figure 2 cancers-13-05590-f002:**
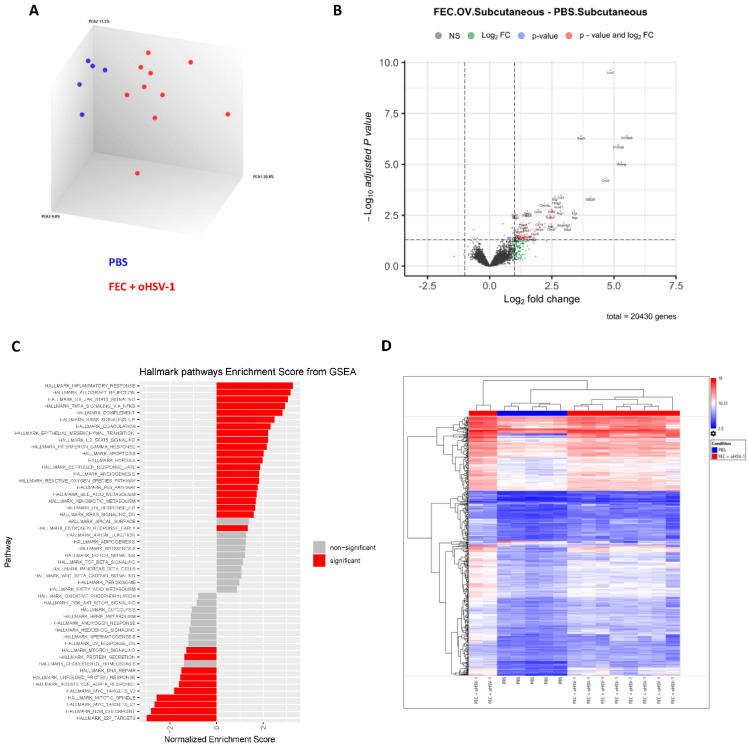
FEC + oHSV-1 therapy upregulates many immune pathways and processes associated with myeloid cells in subcutaneous E0771 tumors; C57/Bl6 mice bearing E0771 subcutaneous tumors were treated with either PBS or FEC + oHSV-1. Tumors were harvested on day 5, and RNA was extracted from whole tumor digests and sent for Clariom analysis. (**A**) A 3-D cluster plot showing the RNA expression correlations between mice treated with PBS (blue; *n* = 5) and FEC + oHSV-1 (red; *n* = 10). (**B**) Volcano plot showing differentially expressed genes between tumors treated with FEC + oHSV-1 and PBS. (**C**) Heat map showing the normalized expression values of genes across all samples. (**D**) Bar plot illustrating the results of hallmark pathway enrichment analysis performed on samples from mice treated with FEC + oHSV-1 compared to those treated with PBS alone.

**Figure 3 cancers-13-05590-f003:**
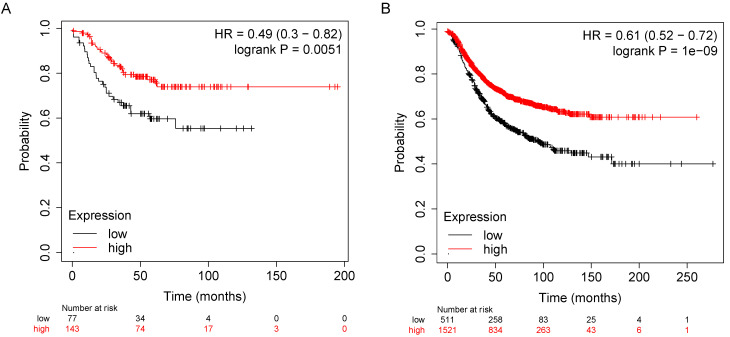
Macrophage gene signature induced by FEC + oHSV-1 therapy correlates with improved clinical outcomes. (**A**) Kaplan–Meier survival plot for TNBC patient RFS (*n* = 220) microarray data based on mean mRNA expression level of the wight macrophage-associated genes in Table 1. (**B**) Kaplan–Meier survival plot for all BC patient RFS (*n* = 2032) and microarray data based on mean mRNA expression level of the eight macrophage-associated genes in Table 1.

**Figure 4 cancers-13-05590-f004:**
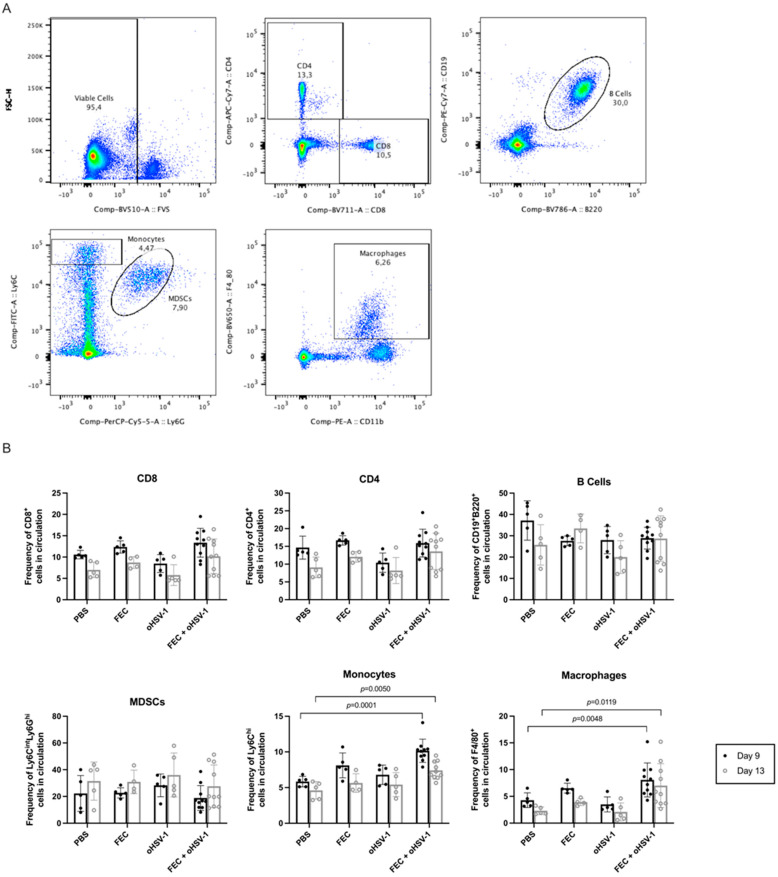
FEC + oHSV-1 therapy increases monocytes and macrophages in the peripheral blood. C57/Bl6 mice bearing E0771 subcutaneous tumors were treated with either PBS (*n* = 5), FEC (*n* = 5), oHSV-1 (*n* = 5), or FEC + oHSV-1 (*n* = 10). Blood was taken 9 and 13 days after the start of treatment and was analyzed via flow cytometry. (**A**) Representative flow plots showing the gating strategy for each immune cell type. (**B**) Bar plots showing the frequencies of T cells (CD8^+^ and CD4^+^), B cells (CD19^+^B220^+^), MDSCs (Ly6C^int^Ly6G^hi^), monocytes (Ly6C^hi^Ly6G^−^), and macrophages (F4/80^+^) in circulating PBMCs. Dots are representative of individual mice. Error bars are representative of the standard deviation. ANOVA test was used for statistical analyses.

**Figure 5 cancers-13-05590-f005:**
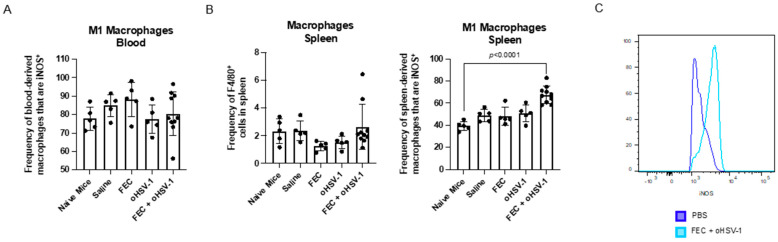
FEC + oHSV-1 therapy increases M1-like macrophages in the spleen. C57/Bl6 mice bearing E0771 subcutaneous tumors were treated with either PBS (*n* = 5), FEC (*n* = 5), oHSV-1 (*n* = 5), or FEC + oHSV-1 (*n* = 10). Mice were sacrificed, and blood and spleens were harvested 10 days after the start of treatment for analysis via flow cytometry. (**A**) Bar plots showing the frequencies of M1 macrophages (CD11b^+^F4/80^+^iNOS^+^ cells) in the peripheral blood. Dots are representative of individual mice. Error bars are representative of the standard deviation. (**B**) Bar plots showing the frequencies of macrophages (CD11b^+^F4/80^+^ cells) and M1 macrophages (CD11b^+^F4/80^+^iNOS^+^ cells) in the spleen. Dots are representative of individual mice. Error bars are representative of the standard deviation. ANOVA test was used for statistical analyses. (**C**) Representative flow histogram showing the expression of iNOS in macrophages from the spleens of mice treated with either PBS or FEC + oHSV-1.

**Figure 6 cancers-13-05590-f006:**
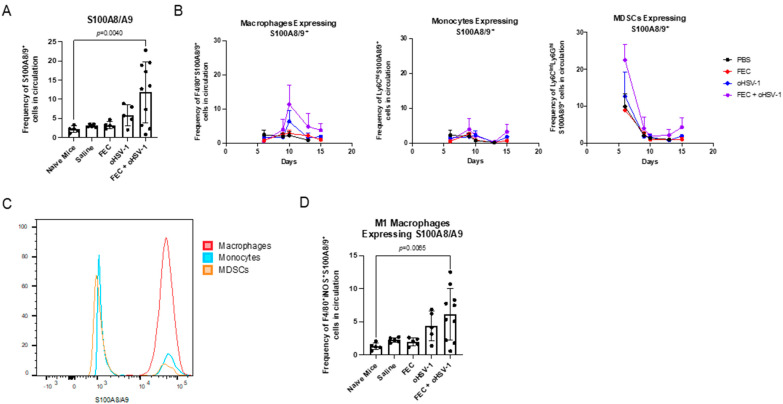
FEC + oHSV-1 therapy increases S100A8A9 in the peripheral blood. C57/Bl6 mice bearing E0771 subcutaneous tumors were treated with either PBS (*n* = 5), FEC (*n* = 5), oHSV-1 (*n* = 5), or FEC + oHSV-1 (*n* = 10). Blood was taken 10 days after the start of treatment for analysis via flow cytometry. (**A**) Bar plots showing the frequencies of S100A8/A9^+^ cells in circulating PBMCs. Dots are representative of individual mice. Error bars are representative of the standard deviation. ANOVA test was used for statistical analysis. (**B**) Line graphs illustrating the mean frequency of macrophages expressing S100A8/A9, monocytes expressing S100A8/A9 and MDSCs expressing S100A8/A9 in the peripheral blood over multiple days. Error bars are representative of standard deviation. (**C**) Representative flow histograms showing the expression of S100A8/A9 on macrophages, monocytes, and MDSCs in the peripheral blood of mice treated with FEC + oHSV-1 therapy. (**D**) Bar plots showing the frequencies of S100A8/A9+ cells in circulating M1-like macrophages. Dots are representative of individual mice. Error bars are representative of the standard deviation. ANOVA test was used for statistical analysis.

**Figure 7 cancers-13-05590-f007:**
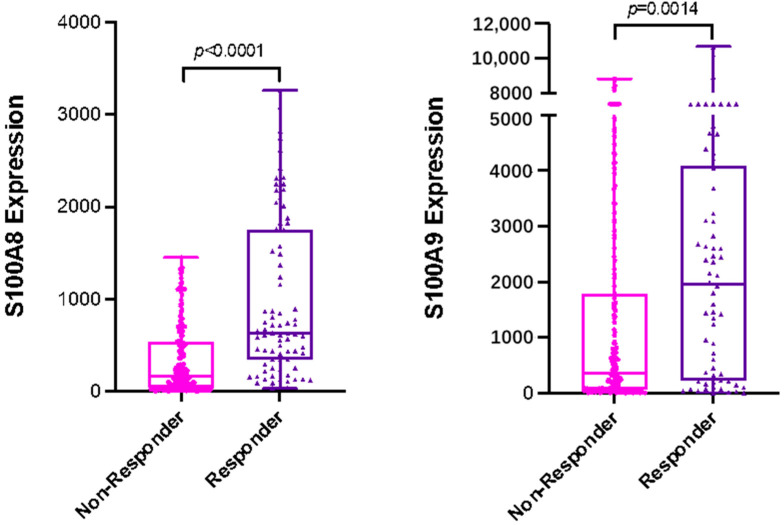
S100A8 and S100A9 are independent predictors of response to FEC therapy in breast cancer patients. S100A8 and S100A9 mRNA expression was compared between responders (*n* = 84) versus non-responders (*n* = 219) to FEC treatment in a cohort of all breast cancer patients (S100A8: AUC = 0.71, *p* < 0.0001; S100A9: AUC = 0.626, *p* = 0.0014). Response defined as complete pathological response versus residual disease after completing therapy.

**Table 1 cancers-13-05590-t001:** Macrophage-associated genes highly upregulated by FEC + oHSV-1 therapy.

Gene.	F + O vs. P(Fold Change)	Function
LCN2	632.83	Regulator of macrophage polarization via STAT3 activation [19] Modulates iNOS and Arg1 in macrophages to promote the anti-inflammatory function [20]
CXCL2	308.84	Secreted by macrophages, regulator of myeloid cell migration
SAA3	180.26	Marker of macrophage infiltration [21] Promotes macrophage differentiation [22]
CXCL3	50.83	Mediator of macrophage recruitment [23]
CCL3	37.62	Macrophage recruitment [24]
IL1A	34.53	Associated with activated macrophages [13]
IL1B	27.81	Associated with activated macrophages [13] Macrophage recruitment [25]
CLEC4E	13.73	Macrophage activation and differentiation [26]

## Data Availability

Clariom S assay data (Figure 2 and Appendix A) can be found in the GEO database (GSE183864).

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
