# Peer review of "Response to FEC Chemotherapy and Oncolytic HSV-1 Is Associated with Macrophage Polarization and Increased Expression of S100A8/A9 in Triple Negative Breast Cancer"

_cancers, 2021, doi:10.3390/cancers13215590_

Round 1
Reviewer 1 Report
The authors of this manuscript are expanding on previous work using combination chemo and oncolytic virotherapy for triple negative breast cancer. In this manuscript the authors seek to dissect the mechanism of myeloid cell regulation and differentiation and identify a therapeutically significant gene signature associated with the switch in myeloid phenotype.
This is an interesting and mostly well written work. It is straight forward, and the results will be of broad interest to those working in the immunotherapy, oncolytic virotherapy fields. Enthusiasm is dampened by a lack of differentiation of this new data from their previously published work. For example, the title of the manuscript under review pretty much restates a conclusion made from their previous paper that myeloid cell populations associated with the therapy were less immunosuppressive. A more descriptive and specific title related to the work under review is suggested. Another criticism might be that the authors are overstating their findings. This can be seen, for example, in the use of the word “driving”. The data only describe associations with successful outcome rather than having proven causality. It may be semantic but the word “driving” can be taken to mean causality.
There are also numerous instances where it is not clear where to find the data described in the results. Simple parenthetical figure references are missing. Additionally, there are missing material and method descriptions (see below).
However, this is an interesting paper that I think will be of interest to many readers. With some redrafting, the paper may be suitable for publication. In addition to the below comments there should be a better referencing of the role of MDSC in the TME in other systems. The authors may have shown this in their model but what about other models? I recommend an expanded discussion section as the present one is quite short.
Specifics:
- Line 67: It is important to clarify subcutaneous TN cells as it could confuse others to think that this is the case for all subcutaneous tumor which is not the case.
- Line 109: “tittered” should read “titered”. This is a common autocorrect issue
- Please review the materials and methods. Several materials and methods are missing. These include: tumor measurement methods and formulas, endpoint description for survival studies, microarray methods. There may be more but this is a good start.
- The sentence in line 159 is not easily understood. There may be a word missing.
- What volume of virus are you using?
- Line 166 “ever” should read “every”.
- Line 166: What is “tumor endpoint”.
- Figure 1D: Why is the FEC + OHSV sample split in the heatmap? It is the same in figure 2.
- Line 230: Where is this data (Figure 4A?B?...)
- Line 237: Please be specific as to where the data can be found. Ibelieve this is Figure 5A,B.
- Section 3.5 doesn’t reference any figure specifically. While it might be clear from the layout it should be referenced specifically. Will the layout be the same in publication?
- Section 3.5: What data set are you using for the human BC data.
- Line 294: I believe this needs a period after the “patients”
- Lines 334-335: Is this in your model or are you making a statement? It should be referenced as this is pretty well known.
- What are the limitations of the whole RNA approach? How can you be sure the genes identified are from MDSC? Do the genes identified play a role in any other cell types? Please discuss this as well as the limitations of your approach.
Author Response
The authors of this manuscript are expanding on previous work using combination chemo and oncolytic virotherapy for triple negative breast cancer. In this manuscript the authors seek to dissect the mechanism of myeloid cell regulation and differentiation and identify a therapeutically significant gene signature associated with the switch in myeloid phenotype.
The authors would like to thank Reviewer #1 for their thorough time reading our manuscript and providing valuable feedback to enhance its readability and impact.
This is an interesting and mostly well written work. It is straight forward, and the results will be of broad interest to those working in the immunotherapy, oncolytic virotherapy fields. Enthusiasm is dampened by a lack of differentiation of this new data from their previously published work. For example, the title of the manuscript under review pretty much restates a conclusion made from their previous paper that myeloid cell populations associated with the therapy were less immunosuppressive. A more descriptive and specific title related to the work under review is suggested. Another criticism might be that the authors are overstating their findings. This can be seen, for example, in the use of the word “driving”. The data only describe associations with successful outcome rather than having proven causality. It may be semantic but the word “driving” can be taken to mean causality.
We agree with this comment and have changed the title to “Response to FEC chemotherapy and oncolytic HSV-1 is associated with macrophage polarization and increased expression of S100A8/A9 in triple negative breast cancer” as a reflection of this.
There are also numerous instances where it is not clear where to find the data described in the results. Simple parenthetical figure references are missing. Additionally, there are missing material and method descriptions (see below).
We have gone through the manuscript and added where appropriate.
However, this is an interesting paper that I think will be of interest to many readers. With some redrafting, the paper may be suitable for publication. In addition to the below comments there should be a better referencing of the role of MDSC in the TME in other systems. The authors may have shown this in their model but what about other models? I recommend an expanded discussion section as the present one is quite short.
We appreciate this comment and have added in more references regarding the role of MDSCs in the TME in both the introduction and discussion sections (lines 62-64, 323-328, 558-566).
Specifics:
- Line 67: It is important to clarify subcutaneous TN cells as it could confuse others to think that this is the case for all subcutaneous tumor which is not the case.
Added/clarified (line 66).
- Line 109: “tittered” should read “titered”. This is a common autocorrect issue
Thank you for bringing this to our attention, change made and checked throughout manuscript.
- Please review the materials and methods. Several materials and methods are missing. These include: tumor measurement methods and formulas, endpoint description for survival studies, microarray methods. There may be more but this is a good start.
We have double checked the materials and methods section and added in any missing information.
- The sentence in line 159 is not easily understood. There may be a word missing.
We have re-worded the sentence for clarification.
- What volume of virus are you using?
We are using 50 µL of virus (line 100).
- Line 166 “ever” should read “every”.
Corrected, thank you.
- Line 166: What is “tumor endpoint”.
We have clarified our tumour endpoint in the text (line 170).
- Figure 1D: Why is the FEC + OHSV sample split in the heatmap? It is the same in figure 2.
In Figure 2D we allowed the heatmap to be clustered both on the gene list as well as the samples. FEC + oHSV-1 samples cluster in two distinct groups, those that are distinctly separate (and believed to be responders to treatment) and those that are more like PBS treated mice (believed to be non-responders to treatment). We have addressed this in the text (line 197-200).
- Line 230: Where is this data (Figure 4A?B?...)
Added.
- Line 237: Please be specific as to where the data can be found. Ibelieve this is Figure 5A,B.
Added.
- Section 3.5 doesn’t reference any figure specifically. While it might be clear from the layout it should be referenced specifically. Will the layout be the same in publication?
We have added in the correct reference for figure 7.
- Section 3.5: What data set are you using for the human BC data.
We are using the NCBI GEO dataset, accrued by the developers of ROCplot.org. This is outlined in the materials and methods (lines 124-138).
- Line 294: I believe this needs a period after the “patients”
Added, thank you.
- Lines 334-335: Is this in your model or are you making a statement? It should be referenced as this is pretty well known.
References added.
- What are the limitations of the whole RNA approach? How can you be sure the genes identified are from MDSC? Do the genes identified play a role in any other cell types? Please discuss this as well as the limitations of your approach.
The authors would like to thank you for this great point you have brought up. We have addressed it in the discussion (lines 376-381).
Reviewer 2 Report
In the manuscript Vito et al. entitled "A combination of clinical chemotherapies and oncolytic HSV-1 2 drives myeloid cell differentiation and polarization to less immunosuppressive phenotypes" authors described the effects of the combination of a chemotherapy and oncolytic virotherapy on a model of murine breast adenocarcinoma. They found that this combined treatment induces a switch of the myeloid infiltrate from an immunosuppressive towards an immunogenic, antitumorigenic, phenotype.
Despite the general interest of the scientific community in finding treatments able to reactivate the immune system in immunosuppressive tumor microenvironment, I think this paper is just only descriptive and speculative and lacks functional experiments demonstrating the data obtained by the transcriptomic analysis.
Moreover, I think that the phenotypic analysis they performed to characterize the immune cells of treated and control animals is poor and superficial. The presence of a single marker on immune cells hardly allows to classify those cells as immunosuppressive or immunostimulatory. A more complete antibodies panel should be used.
Specific issues:
- The fact that 10-20% of mice had a durable response to treatment and 20% of treated mice have a distinct expression profile respect to control animals does not demonstrate a causal effect between the two events since, they used for the microarray analysis a different set of animals respect to those used for the survival experiment. The sentence at page 6 lines 182-185 is therefore just only speculative and should be eliminated or they had to demonstrate the casual relationship between the two results.
- in a previous work (PMID: 34253827) authors found, after a transcriptomic analysis, that the same combination therapy used in the present work induces an upregulation of genes associated with the B-cell lineage. In the present work authors described only the upregulation of genes associated with the myeloid lineage. I think the authors should compare and discuss the new and the previous results also by adding a table showing all the differential expressed genes found in this new analysis.
- on figure 3 they show Kaplan-meier survival plot of TNBC or breast cancer patients based on the mean expression value of 8 genes whose expression seems associated to macrophage cells. They do not indicate whether the expression level of these genes used to calculate the mean values are logarithmic or not. If not, the higher value can have a greater weight, especially if the values are not evenly distributed, and can therefore affect the result on its own. That is, if a gene has a very different expression level respect to control than the other 7 genes, its variation in the expression level could alone sufficiently modify the value of the mean expression of the 8 genes to pass that sample from one category to another despite the other 7 genes do not vary their expression. This type of analysis must then be carried out using logarithmic values and this must be specified in the text.
- Authors mistakenly use the t-test to compare more than 2 groups or to perform multiple pairwise comparisons. In these cases, ANOVA test had to be used since t-test can produce false positive results.
Author Response
In the manuscript Vito et al. entitled "A combination of clinical chemotherapies and oncolytic HSV-1 2 drives myeloid cell differentiation and polarization to less immunosuppressive phenotypes" authors described the effects of the combination of a chemotherapy and oncolytic virotherapy on a model of murine breast adenocarcinoma. They found that this combined treatment induces a switch of the myeloid infiltrate from an immunosuppressive towards an immunogenic, antitumorigenic, phenotype.
Despite the general interest of the scientific community in finding treatments able to reactivate the immune system in immunosuppressive tumor microenvironment, I think this paper is just only descriptive and speculative and lacks functional experiments demonstrating the data obtained by the transcriptomic analysis.
Moreover, I think that the phenotypic analysis they performed to characterize the immune cells of treated and control animals is poor and superficial. The presence of a single marker on immune cells hardly allows to classify those cells as immunosuppressive or immunostimulatory. A more complete antibodies panel should be used.
The authors would like to thank the reviewer for their feedback and have addressed specific concerns below.
Specific issues:
- The fact that 10-20% of mice had a durable response to treatment and 20% of treated mice have a distinct expression profile respect to control animals does not demonstrate a causal effect between the two events since, they used for the microarray analysis a different set of animals respect to those used for the survival experiment. The sentence at page 6 lines 182-185 is therefore just only speculative and should be eliminated or they had to demonstrate the casual relationship between the two results.
We agree with this comment and have amended the sentence to reflect the speculative nature of our claim (lines 197-202).
- in a previous work (PMID: 34253827) authors found, after a transcriptomic analysis, that the same combination therapy used in the present work induces an upregulation of genes associated with the B-cell lineage. In the present work authors described only the upregulation of genes associated with the myeloid lineage. I think the authors should compare and discuss the new and the previous results also by adding a table showing all the differential expressed genes found in this new analysis.
Unfortunately, a table showing the entirety of the differentially expressed genes would be far too large to include in the manuscript, or even in the supplementary data. We have included the dataset in its entirety in the GEO database (GSE183864) and have also discussed and included the overlap of B cell signature with our previous findings (lines 196-200).
- on figure 3 they show Kaplan-meier survival plot of TNBC or breast cancer patients based on the mean expression value of 8 genes whose expression seems associated to macrophage cells. They do not indicate whether the expression level of these genes used to calculate the mean values are logarithmic or not. If not, the higher value can have a greater weight, especially if the values are not evenly distributed, and can therefore affect the result on its own. That is, if a gene has a very different expression level respect to control than the other 7 genes, its variation in the expression level could alone sufficiently modify the value of the mean expression of the 8 genes to pass that sample from one category to another despite the other 7 genes do not vary their expression. This type of analysis must then be carried out using logarithmic values and this must be specified in the text.
For this figure, KMplot.com was used. The details are described in the materials and methods (lines 124-130). In brief, the tool utilized the average expression of the 8 gene signature using logarithmic values. We have clarified in the text.
- Authors mistakenly use the t-test to compare more than 2 groups or to perform multiple pairwise comparisons. In these cases, ANOVA test had to be used since t-test can produce false positive results.
We thank you for bringing this to our attention. We have re-run the statistical analysis for these figures and reported the values using the ANOVA test.
Reviewer 3 Report
The authors of this study applied a combination of chemotherapy with oncolytics to target treatment against a tumor breast cancer line E0771 implanted in mice. The study found tumor progression was effectively delayed in the combination treatment arm, but not in the monotherapy. In addition, a gene expression array experiment was done between FEC + oHSV and PBS treated tumor lines using a Clariom S assay. They found many key alterations in gene expression which included upregulations of a macrophage gene signature, also a myeloid derived cell that plays a primary role in epigenetic reprogramming. Further characterization of this included analysis of blood and spleen by flow cytometry to assess immune subpopulation alterations. The authors found increased monocytes and macrophage subpopulations in samples that underwent FEC + oHSV treatment compared to PBS alone. The study focused on the S100A8 and S100A9 expression in macrophage after results of a Clariom S assay, and characterized the expression between macrophage, monocytes and myeloid derived suppressor cells. The importance of which is related to S100A8/A9 ‘s predictive of response to FEC therapy in breast cancer. Overall the study was well thought out and was well controlled. Based on this, this work merits a favorable outcome for publication with some minor revisions. Here are the suggested edits.
- Fig.1 and Supp Fig.2. What is your N for each group? Please enter it in your figure legends.
- Fig.5A, Line 250. In the figure legend the description does not follow what is shown. Please edit.
- Also no experiment is required, based on the importance of M1 macrophage in the study, can the authors discuss S100A8/9 expression in peripheral blood with M1 subpopulation?
Author Response
The authors of this study applied a combination of chemotherapy with oncolytics to target treatment against a tumor breast cancer line E0771 implanted in mice. The study found tumor progression was effectively delayed in the combination treatment arm, but not in the monotherapy. In addition, a gene expression array experiment was done between FEC + oHSV and PBS treated tumor lines using a Clariom S assay. They found many key alterations in gene expression which included upregulations of a macrophage gene signature, also a myeloid derived cell that plays a primary role in epigenetic reprogramming. Further characterization of this included analysis of blood and spleen by flow cytometry to assess immune subpopulation alterations. The authors found increased monocytes and macrophage subpopulations in samples that underwent FEC + oHSV treatment compared to PBS alone. The study focused on the S100A8 and S100A9 expression in macrophage after results of a Clariom S assay, and characterized the expression between macrophage, monocytes and myeloid derived suppressor cells. The importance of which is related to S100A8/A9 ‘s predictive of response to FEC therapy in breast cancer. Overall the study was well thought out and was well controlled. Based on this, this work merits a favorable outcome for publication with some minor revisions. Here are the suggested edits.
The authors would like to thank Reviewer #3 for their comments and feedback on our manuscript.
- Fig.1 and Supp Fig.2. What is your N for each group? Please enter it in your figure legends.
We have added the n value in the figure legends.
- Fig.5A, Line 250. In the figure legend the description does not follow what is shown. Please edit.
Thank you for bringing this to our attention. The correction has been made.
- Also no experiment is required, based on the importance of M1 macrophage in the study, can the authors discuss S100A8/9 expression in peripheral blood with M1 subpopulation?
The authors would like to thank you for this excellent suggestion. We re-analyzed our flow data and have added this data in figure 6D.
Reviewer 4 Report
The manuscript by Vito et al reports a study in a mouse tumor model of triple negative breast cancer (E0071 cells) showing that that a therapeutic scheme based on the combination of chemotherapy (FEC) and oncolytic virotherapy with oHSV-1 (intratumorally administered) delayed tumor growth and increase survival. Analysis of gene expression on previously aquired RNA-seq data in the same model indicates an increase in inflammatory pathways related to M1 macrophage phenotype. The gene signature emerging from this analysis was analyzed in databsets obtained in TNB cancer patients’ cohorts, showing that in patients, the inflammatory macrophage signature was associated with a better prognosis. In the circulation and spleen of mice bearing TNB tumors and undergoing treatment with FEC and oHSV-1 the author observed an increase in macrophages in particular M1 macrophages (spleen) whereas MDCSs levels were not significantly modified. The treatment (FEC+ oHSV-1) markedly upregulated S100A8 and S100A9 genes and in mouse models with an increase of S100A8/A9 expressing macrophages in a time depended decrease in S100A8/A9 expressing MDSCs. Those authors conclude that the combinatorial therapy modified the TME by decreasing immunosuppressive phenotype in myeloid cells through a mechanism that favors the shift towards the antitumor M1 macrophage phenotype. This process is favored in particular by the oncolytic viral component. To support the importance of inducing Macrophages S100/A8/A9 expressing macrophages by combination therapy to reducing immunosuppression in TME and favor response to treatment, the authors analyze S1000A8 an S100 mRNA expression in a breast cancer patients cohort showing that expression of both mRNAs is associated with a better response to FEC treatment.
The results reported in the manuscript represent a continuation of a recently published study by the same authors showing the positive effect of the combination therapy /FEC +oHSV-1) in improving the efficacy of checkpoint inhibitors in the same tumor model of TNB cancer.
The study adds some information to the previous ones and ascribes the phenomenon to the shift of myeloid cells in TME from immunosuppressing MDCS to antitumoral M1 macrophages.
The manuscript is clearly written, well presented, based on a sound methodology and formally accurate.
There are however few issues:
Major points
1. The overall impact of the results is somehow limited as the main message is that the treatment induced-shift towards M1 macrophages associates with the delay of tumor growth. No causal mechanism is however evidenced thus limiting the significance of the results. Inhibition of S100A8/A9 or M1 macrophage activity (functional inhibition of key molecules or KO mice) could reinforce the significance of the data.
2. As stated by the authors the role of S100A8/A9 un cancer progression is controversial and multiple evidence point towards a protumorigenic role (Jo, S.H., Heo, W.H., Son, HY. et al. S100A8/A9 mediate the reprograming of normal mammary epithelial cells induced by dynamic cell–cell interactions with adjacent breast cancer cells. Sci Rep 11, 1337 (2021). https://doi.org/10.1038/s41598-020-80625-2 ). This aspect should therefore be clarified and analyzed more in depth.
3. The association of (S100A8 and S100 a) with the response to chemotherapy could mean that extended cell damage and stress could favor the expression of these molecules.
4. Overall the mechanistic explanation of the putative role of S100A8/A9 in the phenotype shift from myeloid suppressors (TAM or MDSCs) to M1 macrophages is not strongly supported.
Minor points:
The last part of the discussion is focused on TIL and CP blockade rather than on the complex mechanism of tumor immunosuppression and its modulation by the combination therapy.
-Line 324 “STAT-3 mediates suppression of myeloid cells”. Probably there is a missing word
-Legend to figure 6: “(c) mean expression of macrophages e, monocytes…expressing S100A8/A9...” Probably the authors meant “mean frequency of macrophages expressing...”.
Author Response
The manuscript by Vito et al reports a study in a mouse tumor model of triple negative breast cancer (E0071 cells) showing that that a therapeutic scheme based on the combination of chemotherapy (FEC) and oncolytic virotherapy with oHSV-1 (intratumorally administered) delayed tumor growth and increase survival. Analysis of gene expression on previously acquired RNA-seq data in the same model indicates an increase in inflammatory pathways related to M1 macrophage phenotype. The gene signature emerging from this analysis was analyzed in datasets obtained in TNB cancer patients’ cohorts, showing that in patients, the inflammatory macrophage signature was associated with a better prognosis. In the circulation and spleen of mice bearing TNB tumors and undergoing treatment with FEC and oHSV-1 the author observed an increase in macrophages, in particular M1 macrophages (spleen), whereas MDCSs levels were not significantly modified. The treatment (FEC+ oHSV-1) markedly upregulated S100A8 and S100A9 genes and in mouse models with an increase of S100A8/A9 expressing macrophages in a time dependent decrease in S100A8/A9 expressing MDSCs. Those authors conclude that the combinatorial therapy modified the TME by decreasing immunosuppressive phenotype in myeloid cells through a mechanism that favors the shift towards the antitumor M1 macrophage phenotype. This process is favored in particular by the oncolytic viral component. To support the importance of inducing Macrophages S100/A8/A9 expressing macrophages by combination therapy to reducing immunosuppression in TME and favor response to treatment, the authors analyze S1000A8 and S100 mRNA expression in a breast cancer patients cohort showing that expression of both mRNAs is associated with a better response to FEC treatment.
The results reported in the manuscript represent a continuation of a recently published study by the same authors showing the positive effect of the combination therapy (FEC +oHSV-1) in improving the efficacy of checkpoint inhibitors in the same tumor model of TNB cancer.
The study adds some information to the previous ones and ascribes the phenomenon to the shift of myeloid cells in TME from immunosuppressing MDCS to antitumoral M1 macrophages.
The manuscript is clearly written, well presented, based on a sound methodology and formally accurate.
The authors would like to thank Reviewer #4 for their feedback on our manuscript and their suggestions, which have aided us in improving our work.
There are however few issues:
Major points
- The overall impact of the results is somehow limited as the main message is that the treatment induced a shift towards M1 macrophages associates with the delay of tumor growth. No causal mechanism is however evidenced thus limiting the significance of the results. Inhibition of S100A8/A9 or M1 macrophage activity (functional inhibition of key molecules or KO mice) could reinforce the significance of the data.
The authors agree with this important statement from Reviewer #4. As seen in the literature, there is much controversy over classifying macrophages under the classical terms of M1 and M2. For this reason, we tried to use the term “M1-like” in most cases and clearly state that we were basing this classification on iNOS expression. We were interested in showing causal mechanism with functional in vivo studies but were unable to find suitable antibodies for inhibition of S100A8/A9. Additionally, as these proteins are multi-functional it is difficult to assess their specific role in anti- or pro-tumorigenesis with KO mice, as it also results in changes to other important signaling pathways as well. In future work, we hope to work on indirectly inhibiting the S100A8/A9 function via TLR4 inhibition to begin assessing the exact mechanistic link, though we feel this falls outside the realm of this manuscript.
- As stated by the authors the role of S100A8/A9 in cancer progression is controversial and multiple evidence points towards a protumorigenic role (Jo, S.H., Heo, W.H., Son, HY. et al.S100A8/A9 mediate the reprograming of normal mammary epithelial cells induced by dynamic cell–cell interactions with adjacent breast cancer cells. Sci Rep11, 1337 (2021). https://doi.org/10.1038/s41598-020-80625-2 ). This aspect should therefore be clarified and analyzed more in depth.
We have added additional discussion around the conflicting literature regarding anti- or pro-tumorigenic functions of S100A8/A9 to our discussion (lines 398-403).
- The association of (S100A8 and S100 a) with the response to chemotherapy could mean that extended cell damage and stress could favor the expression of these molecules.
This is an interesting point. We have added in a portion to the discussion about this (lines 403-406).
- Overall the mechanistic explanation of the putative role of S100A8/A9 in the phenotype shift from myeloid suppressors (TAM or MDSCs) to M1 macrophages is not strongly supported.
We agree with Reviewer #4 that we have not fully elucidated any mechanistic link between S100A8/A9 and macrophage polarization. We have gone through the manuscript and carefully edited our title and semantics to reflect that our findings are an “association”, and not a causal mechanistic link.
Minor points:
The last part of the discussion is focused on TIL and CP blockade rather than on the complex mechanism of tumor immunosuppression and its modulation by the combination therapy.
We have changed this and added more detail to the discussion (lines 385-403).
-Line 324 “STAT-3 mediates suppression of myeloid cells”. Probably there is a missing word
There is no missing word in this sentence. It is “STAT3-mediated”, not mediates.
-Legend to figure 6: “(c) mean expression of macrophages e, monocytes…expressing S100A8/A9...” Probably the authors meant “mean frequency of macrophages expressing...”.
Corrected (line 286).
Round 2
Reviewer 2 Report
I noticed that the authors clarify all the “specific issues” I highlighted, but I think they missed the major points reported in the first part of my previous comments, that is the fact that:
-I think the paper is basically descriptive and lacks functional experiments demonstrating the data obtained by the transcriptomic and flow cytometry analyses. For example, is the M1 shift of the macrophage population essential to the response to therapy they found in mice treated with FEC + oHSV-1? that is, what happen using mice depleted in the macrophage population? (I understand what they reply to the reviewer 4 that asked my same question but I think that in the present form the work could not have a big impact on the scientific community)
- I think that a deeper phenotypic analysis of the immune cells had to be performed. For example, even if Ly6ChiLy6G-cells were often considered inflammatory monocytes, there are reports describing these cells as MDSC immunosuppressive cells (doi.org/10.1016/j.cellimm.2021.104300, doi.org/10.1128/IAI.01590-13). Moreover, authors quantify the level of macrophages in the blood without characterizing their M1 versus M2 phenotype (They analyzed iNOS only in the spleen). It is therefore difficult to evaluate whether their increase in the blood is due to an increase in the M1 subtype.
I therefore think that the impact of the work remains limited, even if various improvements have been made.
Author Response
Reviewer #2
I noticed that the authors clarify all the “specific issues” I highlighted, but I think they missed the major points reported in the first part of my previous comments, that is the fact that:
The authors would like to thank Reviewer #2 for their input to address the impact and significance of our work.
-I think the paper is basically descriptive and lacks functional experiments demonstrating the data obtained by the transcriptomic and flow cytometry analyses. For example, is the M1 shift of the macrophage population essential to the response to therapy they found in mice treated with FEC + oHSV-1? that is, what happen using mice depleted in the macrophage population? (I understand what they reply to the reviewer 4 that asked my same question but I think that in the present form the work could not have a big impact on the scientific community)
The authors would like to thank the reviewer for bringing up this crucial point, highlighting the potential impact (or lack thereof) of our work to the broader scientific community. We understand that the manuscript lacks the functional experiments needing to show the mechanistic link between macrophage polarization and therapeutic efficacy. However, we believe that this level of functional immune analyses is outside the scope of this manuscript and journal.
We have previously shown that FEC + oHSV-1 therapy is driven by B cells, particularly those of a memory phenotype (https://doi.org/10.1038/s42003-021-02375-9). We believe that polarization of macrophages to an M1 phenotype is a result of increased B cells into the TME. We have added in this correlation to our introduction (lines 73-75) and discussion (lines 371-372).
- I think that a deeper phenotypic analysis of the immune cells had to be performed. For example, even if Ly6ChiLy6G-cells were often considered inflammatory monocytes, there are reports describing these cells as MDSC immunosuppressive cells (doi.org/10.1016/j.cellimm.2021.104300, doi.org/10.1128/IAI.01590-13). Moreover, authors quantify the level of macrophages in the blood without characterizing their M1 versus M2 phenotype (They analyzed iNOS only in the spleen). It is therefore difficult to evaluate whether their increase in the blood is due to an increase in the M1 subtype.
I therefore think that the impact of the work remains limited, even if various improvements have been made.
The authors agree that classification of inflammatory monocytes and MDSCs in mice is somewhat controversial in the literature, with various research groups reporting different subsets by different names. While Tam and colleagues (doi.org/10.1128/IAI.01590-13) have reported Ly6ChiLy6G- as MDSCs, they have acknowledged that this is a transient cell state and that these cells do further undergo differentiation to become macrophages. Contrary to this finding, the group of cells classically determined to be “myeloid-derived suppressor cells” are named so for their stagnant immature state, as they do not fully differentiate to a definitive cell type. Indeed, in our murine model, we have previously shown that this cell type (Ly6CintLy6G+) is strongly associated with a lack of therapeutic efficacy, tumor growth and positive association with various immunosuppressive markers commonly found on MDSCs (https://doi.org/10.1038/s42003-021-02375-9).
We do however agree with the reviewer’s comment that it is difficult to evaluate whether the increase in macrophages in the blood is due to an increase in the M1 subtype, so we have gone back to our data and done this analysis. You will see it added into the manuscript in Figure 5a (lines 260-262, 278-281). We have also addressed these findings in the discussion (lines 406-409).
Round 3
Reviewer 2 Report
no more comments